# NETosis in Wound Healing: When Enough Is Enough

**DOI:** 10.3390/cells10030494

**Published:** 2021-02-25

**Authors:** Maurizio Sabbatini, Valeria Magnelli, Filippo Renò

**Affiliations:** 1Department of Science and Innovation Technology (DISIT), Università del Piemonte Orientale—via T. Michel 11, 15121 Alessandria, Italy; maurizio.sabbatini@uniupo.it (M.S.); valeria.magnelli@uniupo.it (V.M.); 2Innovative Research Laboratory for Wound Healing, Health Sciences Department, Università del Piemonte Orientale, via Solaroli 17, 28100 Novara, Italy

**Keywords:** netosis, wound healing, inflammation, innate immunity

## Abstract

The neutrophils extracellular traps (NETs) are a meshwork of chromatin, histonic and non-histonic proteins, and microbicidal agents spread outside the cell by a series of nuclear and cytoplasmic events, collectively called NETosis. NETosis, initially only considered a defensive/apoptotic mechanism, is now considered an extreme defensive solution, which in particular situations induces strong negative effects on tissue physiology, causing or exacerbating pathologies as recently shown in NETs-mediated organ damage in COVID-19 patients. The positive effects of NETs on wound healing have been linked to their antimicrobial activity, while the negative effects appear to be more common in a plethora of pathological conditions (such as diabetes) and linked to a NETosis upregulation. Recent evidence suggests there are other positive physiological NETs effects on wound healing that are worthy of a broader research effort.

## 1. Introduction

Neutrophils, also known as neutrophilic granulocytes or polymorphonuclear leukocytes (PMNs), are the most abundant white blood cells in the human circulatory system. They play a crucial role in the innate immune defense against bacteria, fungi, and viruses, and they are potentially harmful to the host as well.

Neutrophils are recognized as highly versatile and sophisticated cells, able to greatly extend their lifespan depending on their activation status, and cross-talk with other inflammatory cells. It is now thought that the circulatory half-life of neutrophils is longer than previously estimated (several days as opposed to several hours) [1].

Their activation and microbicidal activity are strictly controlled by a plethora of stimuli, and recent evidence suggests they are quite versatile and can perform previously unsuspected functions, such as reverse transmigration, crosstalk, and regulation of other leukocyte populations [1].

The antimicrobial and cytotoxic action mechanisms of neutrophils consist of phagocytosis, generation of reactive oxygen species (ROS), and the degranulation of several microbicidal factors such as α-defensins, cathelicidin, elastase, cathepsin G, and lactoferrin. Neutrophils also exhibit a remarkable de novo biosynthetic capacity for C-X-C and C-C chemokines; proinflammatory, anti-inflammatory, and immunoregulatory cytokines; as well as angiogenic and fibrogenic factors and matrix metalloproteinases [1,2,3,4].

In addition to the numerous actions that characterize the defensive response of neutrophils, another particular way of entrapping and killing pathogens has been observed. In 2004, Brinkmann et al. [5] reported the extrusion by neutrophils in a meshwork of chromatin fibers decorated with granule-derived antimicrobial peptides and enzymes capable of killing Gram-positive and Gram-negative bacteria. Due to the modality of defensive action, this defensive meshwork has received the denomination of neutrophil extracellular traps (NETs). NETs were also observed to be efficient in the host defense against fungi [6], although not necessary against enveloped viruses [7].

NETs are composed of highly decondensed chromatin fibers, having a diameter of 15 to 17 nm, derived from nuclear components accompanied by histone proteins and complexed with microbicidal globular proteins, such as elastase, cathepsin G, and myeloperoxidase, which are normally stored in neutrophil granules [5,8].

NETs are released in the extracellular space where the chromatin meshwork entraps microbes, limiting their diffusion and concentrating the neutrophil factors, and thus increasing the microbicidal effects [9]. NET releases are fundamental as a defensive mechanism, when the size of the pathogens makes phagocytosis an unreliable process [10].

A further mechanism through which NETs carry out their microbicidal activity is related to the ability of DNA to induce chelation of manganese and other ions. In particular, manganese plays an important role in the proliferation and survival of microbes. Together with other divalent cations, it is used for the transport of electrons across the cellular membrane, a fundamental process used by bacteria to obtain chemical energy for their vital activity [11]. As a consequence of the chelating activity of the DNA in the meshwork of NETs, decreased ion transport impedes the survival of microbes [12,13]. Further, it has been observed that NETs not only have an active function in eliminating pathogens, but also regulate the local inflammatory process [14,15].

Initially, NET formation was interpreted as a particular form of cell death, different from apoptosis or necrosis (no caspases/DNA fragmentation or necrosis death signals activation occur) [15], responsible for terminating the short life of activated neutrophils during an explosive event producing microbiocidal effects. Following this interpretation, the process was called NETosis [16].

However, the identification of NET formation stages has recently led researchers to revise the death concept of NETosis (Figure 1). The formation of NETs starts from the dissolution of the nuclear envelope, freeing decondensed chromatin threads into cytoplasm. Subsequently, the granular membranes also disappear, allowing the mixing of nuclear and granular components; throughout this process, the cell membrane is intact. In the final step, chromatin threads and granular components are released by a process that preserves the integrity of the cytoplasmic membrane of polymorphonuclear cells (PMNs) [15]. Although enucleated, the remnants of the neutrophils continue their defensive antimicrobial performance by active phagocytosis, albeit for only a few hours [14,15,17].

This last observation led to the hypothesis that this process cannot be properly understood as a cell death process, because neutrophils are still alive after NET extrusion. In these terms, the NETosis process is no different than other defensive bursting effects, such as those using ROS, which in addition to their efficient microbicidal effects, result in the death of immune cells [18].

From this point of view, some researchers have criticized the use of the term NETosis, preferring the term NET formation [19]. NET formation appears to be a widespread phenomenon within immune cell responses, and in addition to neutrophils, other immune cells have been reported to develop the process of NET formation [20,21,22,23], so that a broader term extracellular trap (ET)osis has been proposed [24].

Neutrophils, like monocytes, work to maintain the integrity and health of the organism. They control the molecular patterns of microbes, known as pathogen-associated molecular patterns (PAMPs), and/or molecules from damaged cells of host origin, known as damage-associated molecular patterns (DAMPs), using pattern recognition receptors (PRRs). Activation of the immune system response is mediated by the detection of PAMPs and/or DAMPs. PAMPs are a molecular structure; a component of the cell wall, such as lipopolysaccharide (LPS), peptidoglycan, lipoteichoic acids, and cell-wall lipoproteins; flagellar components such as flagellin; or β-glucan, which is a component of fungal cell wall [25].

DAMPs are molecules normally produced by cells, and in healthy conditions they remain inside the cells. When DAMPs are detected outside the cells, due to stress or injury, they can trigger a strong reaction from the immune system. Typical DAMPs include nuclear proteins, histones, ATP, mitochondrial components such as mitochondrial DNA, and uric acid [26,27]. When DAMPs are detected by PRRs such as toll-like receptors (TLR) there is an activation of the immune system response. This clears dead cells, which triggers inflammation and is necessary for tissue regeneration [26,28].

The formation of NET begins with the activation of neutrophils through the recognition of stimuli, leading them to package and activate the NADPH oxidase (NOX) complex through protein kinase C (PKC)/Raf/MERK/ERK, as well as increase cytosolic Ca^2+^ [29,30].

Bacteria can induce NET production by activating TLR4 [22,31]. Stimulation of TLR4 activates a pathway with as its principal intermediates NADPH oxidase 2 (NOX2), an enzyme that generates reactive oxygens species (ROS), myeloperoxidase enzyme (MPO), and peptidylarginine deaminase 4 (PAD4) [32]. This process takes from 1 h to 4 h to be completed after the initial stimulus.

During NET formation, the separation of chromatin into eu- and hetero-chromatin is lost [33]. In this process, the enzymes of azurophilic granulocyte, elastase, and myeloperoxidase are involved, which move into the nucleus in the early stage of NETosis in an unknown way. Elastase is the first to enter the nucleus, where it determines the cleavage of the histone linker H1, and modifies the histone core [33]. The elastase action makes it the first fundamental limiting factor in the formation of the traps; mice deficient in this enzyme cannot produce NETs [33]. Later, MPO enters the nucleus, increasing the decondensation of chromatin, likely by hypochlorite synthesis; subjects with an altered MPO gene cannot form NETs [29,34].

Another enzyme involved in this pathway is peptidylarginine deiminase 4 (PAD4), which induces the deamination of the arginine residues to citrulline in the histone 3 and 4 (histone citrullination or demethylimination), producing a weaker binding to DNA and chromatin decondensation [8,29,30,34]. The role of PAD4 in NET formation has been studied in knock-out mice that were unable to form NETs [8,29,35,36]. Finally, the pore-forming protein gasdermin D has been proven to be involved in NETosis, allowing for DNA and associated protein extrusion [30,37,38].

Recently, it was demonstrated that NADPH oxidase-dependent autophagy is involved in NETs [39]. Neutrophil stimulation by phorbol myristate acetate (PMA) produces a giant vacuole similar to autophagosome [40,41]. The cytoskeleton is also involved in the regulation of NET formation, as the formation of tubules can direct the movement of granules during exocytosis and phagocytosis [42].

A particular short NET formation pathway has been recently observed, where mitochondrial DNA is released instead of nuclear DNA, depending on ROS formation. This quick process occurs in 80% of neutrophils within 15 min, following C5a or LPS recognition [43].

## 2. The Dark Side of NETosis

Neutrophils are mostly seen as playing a beneficial role to the host, but their improper activation can also lead to tissue damage during an autoimmune or exaggerated inflammatory reaction [44,45,46,47] (Figure 2).

Neutrophils are involved in acute infections and inflammation. Excessive activation of neutrophils may lead to the development of multiple organ dysfunction syndrome, where the lungs are the main target, such as acute lung injury (ALI), and in its more severe form, acute respiratory distress syndrome (ARDS) [48].

NETs act by promoting differentiation of pulmonary fibroblasts into active myofibroblasts responsible to fibrotic effect, and NETs in close proximity to alpha-smooth muscle actin (α-SMA)-expressing fibroblasts were found in biopsies from patients with fibrotic interstitial lung disease [49].

NETs have been implicated in chronic inflammatory disorders or ageing-related diseases, such as atherosclerosis, psoriasis, rheumatoid arthritis, inflammatory bowel disease, diabetes, and cancer [36].

In cancer, the double face of NET formation is evident, since NETs show antitumor activity but can stimulate tumor invasiveness [50], and protect tumor cells from cytotoxic lymphocytes activity [51]. In particular, in a murine breast cancer model, NETs promote the progression of tumor metastasis, due to protein factors strictly associated with the DNA mesh. The presence of NETs was observed around metastatic cells, and DNAse activity blocked the cell invasiveness [52]. Notably, the quantification of NET formation has been suggested as a prognostic biomarker in neoplastic disorder [52,53,54,55], or a potential target for new therapeutic approaches using DNAse I to disrupt the effect of the DNA-mesh or inhibitors of PAD4 [52,56].

In a diabetic patient, neutrophils showed an increase in spontaneous NETosis that induced an important wound healing alteration [57].

In systemic lupus erythematosus, NETs have been claimed to be responsible for the autoimmune response [37,58,59]. In an experimental lupus-like autoimmunity model in a mouse, aberrant NET formation and consequent uncontrolled release of inflammatory mediators occurred. These events can induce a delay in the macrophage dependent downregulation of inflammation, causing an inappropriate persistence of autoantigen [60] responsible for the aggravated condition in systemic lupus erythematosus [58].

In particular, the use of chromatin elements in forming NETs have raised the question as to whether NET formation is the cause of developing several forms of autoimmunity, because hide antigens become accessible to immune cells surveillance [61].

Macrophages are involved in removing NETs: M1 (pro-inflammatory type) macrophages, and M2 (anti-inflammatory and tissue remodeling type) macrophages are equally involved in tissue clearance [62]. Delayed macrophages-dependent clearance could be responsible for the persistence of autoimmunity triggering factors [60,63].

NETs DNA and histones activate the platelets and the coagulation cascade and NETs form aggregates called AggNET, which are scaffolds for erythrocytes and activated platelets. At the same time, elastase inactivates the main coagulation inhibitors antithrombin III and tissue factor pathway inhibitor (TFPI), and a further formation of thrombi in the blood vessels occurs with consequent damage to the lungs, heart, and kidneys [64].

Molecules that are able to counteract the production of NETs by neutrophils have already been reported [65,66]. Furthermore, given the involvement of neutrophils in coagulation, it is worth evaluating the activity of neutrophils and the formation of NETs in patients with congenital (e.g., hemophilia A) or acquired (e.g., disseminated intravascular coagulation) coagulopathies. Hence, according to Barnes et al. and Tomar et al., the development of novel therapeutic strategies targeting neutrophils, such as inhibitors of neutrophil recruitment or NET formation, could help reduce thrombosis and mortality in COVID-19 patients [67,68], as well as circulatory complications in infections caused by other pathogens.

Recently, exacerbated neutrophils response and NET production have been suggested as being involved in COVID-19-associated pneumonitis and/or ARDS [67,69], since excessive NET production also causes the acute cardiac and kidney injuries common in patients with severe COVID-19 [70].

In COVID-19, an uncontrollable and progressive inflammation, due to a cytokine storm, is caused by the alteration in the crosstalk between macrophages and neutrophils [67]. Neutrophilia predicts poor outcomes in patients with COVID-19 [71].

In conclusion, excessive NET formation affects the inflammatory response, worsening tissue damage in pathologic conditions.

## 3. The Bright Side of NETosis

In addition to the NET formation being a process involved in defensive mechanisms against microbes, several authors have shown that NETs play a role in resolving inflammation, as demonstrated for gouty arthritis [72], even if there are not enough to fully resolve gouty arthritis [73,74].

In the case of neutrophils, recruitment to the area of inflammation induces the release of a particular form of NETs called aggregated NETs, which modulate inflammation by binding and sequestering inflammatory cytokines such as IL-1β and IL-6, which are then degraded by the serine proteases attached to their meshwork [75].

Therefore, NETs also act as regulators in the inflammatory process when they can act as a key component in the initiation and resolution of inflammation. The NETosis effect as a pro- or anti-inflammatory agent depends on the quantity, quality, and length of the NETs, highlighting new sophisticated and complex mechanisms adopted by neutrophils to play their role in immune defense and inflammation [36]. Nevertheless, the variety of experimental conditions adopted in the study of NET formation still did not show a clear correlation between NET effects and their quantity/quality [75,76].

When summarizing information about the dark and light sides of neutrophils NETs, we are faced with a mechanism that increases the microbicidal properties of neutrophils and regulates the inflammatory response. Neutrophils balance their pro- and anti-inflammatory action, but in the presence of conditions that exacerbate NET formation, they become pathogenic agents [30,36].

## 4. NETs Formation in Wound Healing

In wound healing, the Janus-faced behavior of NETs is not well-documented (Figure 3). In particular, increased NETosis has been shown to impair wound healing in a plethora of pathological conditions [77], including in skin tumorigenesis [78]. Furthermore, delayed wound healing due to NET formation has been well-reported in diabetic patients [46,79].

In these patients, an increased PAD4-mediated NET formation has been associated with increased release of elastase, responsible for extracellular matrix degradation and delayed wound healing [80,81,82]. Additionally, in *Padi4*^−/−^ diabetic mice, wound healing was accelerated [83].

The importance of excessive NET production in relation to disease manifestation or progression has gained further evidence in the efficacy of metformin diabetic treatment. Metformin, a well-known first-line drug for the treatment of type 2 diabetes, in addition to its incompletely understood glucose-lowering mechanism, has revealed the ability to reduce the concentrations of NET components by the inhibitory effect on the PKC-NADPH oxidase pathway, responsible of the first metabolic steps leading to NET production [84]. Again, these findings confirm that the observed tissue damage induced by NET is a reflex of an altered production, and not a manifestation of a harmful defensive mechanism *in sè*.

NETosis dysregulation can also be induced by chronic inflammation, a mechanism responsible for the delayed wound healing process, as demonstrated in diabetic foot ulcer (DFU) patients, where NOD-like receptor protein (NLRP)-3 inflammasome-NETs axis was upregulated compared with both controls and diabetic patients with no ulcer [85,86]. Therefore, NETs have been recognized as markers of wound healing impairment in diabetic foot ulcers patients [87].

The chronic inflammation-inducing NETosis dysregulation model is well-represented in psoriasis. Psoriasis pathogenesis depends on IL-17 and IL-23 levels, and drugs targeting these cytokines are used to control the disease [88]. The efficacy of these treatments has been attributed to the block of a particular subset of T cells (called Th17) in producing IL-17 and in blocking their expansion induced by IL-23, IL-21, and IL-1 β [89,90,91,92]. However, mast cells and neutrophils are the main producers of IL-17 in the skin, and in psoriasis. In particular, IL-17 is released during the formation of extracellular traps by neutrophils, whereas IL-23 and IL-1β are involved in degranulation and extracellular trap formation in mast cells [22,37].

Another process mediated by IL-17 has been reported by Frangou et al. NET scaffolds decorated with tissue factor (TF) and interleukin-17A (IL-17A) are involved in thromboinflammation and lung fibrosis in systemic lupus erythematosus (SLE) patients [93]. IL-17A is a proinflammatory cytokine that was shown to be associated with an increase and propagation in a fibrotic process in several different tissues, including lung, skin, liver, and others, with the exception of the kidney [94]. In SLE patients, NETs derived from an impaired autophagic mechanism are enriched with TF and IL-17A, which both work as active proteins: TF induces thrombin generation, and IL-17A behaves as promoter of collagen deposition [93].

In addition to the effects of NETs on the extracellular matrix and on the immune response, their direct effects on the cell populations involved in wound healing were observed. For example, NETs in ALI/ARDS inflammation induced M1 pro-inflammatory macrophage polarization [48], whereas in diabetic wounds, NETs upregulated NLRP3 and pro-IL-1β levels via the TLR-4/TLR-9/NF-κB signaling pathway in macrophages, sustaining a local pronged inflammatory response [85].

NETs increased the expression of connective tissue growth factors, collagen production and proliferation/migration in fibroblasts expressing alpha-smooth muscle actin (α-SMA) [49], and were detected in close proximity to this cell population in biopsies from patients with skin scar tissue [49].

Notably, the effects of NETs on endothelial cells (ECs) and angiogenesis during wound healing have not been extensively explored. The only information we have on the interaction between endothelial cells and NETs is from studies on the pathophysiology of atherosclerosis [95] or other vascular diseases. High levels of NETs induced endothelial cells death in vasculitis [96] and endothelial-to-mesenchymal transition in lupus nephritis [97]. However, NETs were able to induce proliferation, destabilize intercellular junctions, and increase cell motility in ECs, along with in vitro angiogenesis via TLR4 [98]. Interestingly, NET concentrations capable of stimulating angiogenesis were not measured.

In wound healing, keratinocytes play a pivotal role during the initial defensive process, recruiting neutrophils, macrophages, and other leukocytes, and they are essential in the repairing process. Keratinocytes express different kinds of TLR on their cell surface [99,100], with TLR-4 appearing to be involved in acute wound healing, as TLR4 blockade delays the migration of normal primary human epidermal keratinocytes (NHEK) and abolishes the phosphorylation of p38 and JNK/MAPK, and IL-1β production [101]. External TLRs 1, 2, 4, 5, and 6 are found on the cell surface, and recognize ligands of mainly bacterial and fungal origin. Internal members of TLR family (TLR 3, 7, 8, and 9) are located in endosomes where their primary function is to detect microbial and host-derived nucleic acids [100]. Recently, endosomal TLRs have been shown to be potential contributors to inflammation in different human studies and rodent models [102,103].

TLR9 recognizes and binds unmethylated CpG residues in the DNA uptaken by endocytosis [104], and TLR9 ectodomain is subsequently cleaved to produce a functional receptor for the recruitment of adapter molecules and activation of NF-κB or interferon (IFN) [103]. Proteolytic cleavage of TLR9 is a prerequisite for its activation, since after cleavage, one of its fragments binds with Myd88 to induce downstream signaling in a wide variety of cells [103]. TLR9 is expressed by subsets of B cells and dendritic cells in humans, and is involved in local as well as systemic inflammation [105]. Moreover, the TLR-9 receptor is involved in double-strand DNA internalization through vesicular uptake in macrophages [106]. Keratinocytes express TLR9, with its activation inducing type I INF [100]. In psoriasis, an increase in TLR9 response induces the production of a greater amounts of IFN-β characterizing the psoriatic lesion [107].

NETs can also exert a stimulating positive role on the wound healing process, even if it is only now beginning to be investigated. NETs act on keratinocytes through the internalization of double-strand DNA by TLR9 receptors, which induce a NF-kB-dependent keratinocyte proliferation [108]. This phenomenon is NET-concentration-dependent, as low and high NETs concentrations induce an opposite effect on in vitro wound healing [108], with low physiological NETs concentration increasing keratinocyte proliferation. This NET concentration-dependent effect on in vitro keratinocytes proliferation and wound healing was strongly reduced in elderly subjects (over 65 years old). Moreover, LPS-stimulated neutrophils from elderly subjects produced a higher NET concentration compared to adult subjects (20–40 years old), but NETs were less effective in inducing bacterial toxicity and keratinocyte proliferation (Sabbatini et al., unpublished results).

These findings are in contrast with the results obtained by Tseng et al. in a murine model of *Staphylococcus aureus* infection, where less NET production was observed in neutrophils obtained from elderly animals compared to those obtained from younger animals [109], while in humans, an age-associated reduction in IL-8 and LPS-induced NET formation has been observed [110].

The difference in the experimental model and neutrophils stimulation could explain, at least in part, the different findings. Moreover, the NETs from elderly human subjects had larger DNA fragments compared to NET production of young subjects (Sabbatini et al., unpublished results) and large DNA fragments have less efficiency in ion chelation, a key quality for microbicidal performance [111].

Furthermore, larger DNA fragments are less efficient in the interaction and activation of TLR9 [112], which explains the absence of keratinocyte NETs induced proliferation. Finally, in a murine model, the activity/NETosis of PAD4 increased with age, and PAD4/NETs has been implicated in age-related organ fibrosis [80].

The Janus-face of NET production can also be appreciated when comparing the inefficient antimicrobic defense and the wound healing process observed in diabetic patients.

Type 2 diabetes mellitus (T2D) patients are known to be at increased risk for infections and wound healing impairment, which have been partially linked to a delayed release of short-living NETs [57,113]. T2D patients exhibit NETs decorated with a low level of cathelicidin (LL-37), which possess poor antimicrobial action, probably due to scaffold defects in extracellular NET threads, induced by constant hyperglycemia.

When patients are treated with the macrolide clarithromycin, the antibacterial ability is restored by up-regulation of LL-37 on NETs [114]. At the same time, improved wound healing was observed due to dermal fibroblasts activation and differentiation [114]. These findings indicate a direct positive physiological role of NETs in wound healing, beyond their defensive antimicrobic action. However, the physiological importance of NET in wound healing remains inadequately investigated.

## 5. Concluding Remarks

NET formation is a neutrophils physiological response that stems the tissue invasion by external agents and regulates inflammation.

This process is highly resolutive and needs the balanced clearance action of macrophage, but higher levels of spontaneous or induced NETosis can unveil NETs’ negative effects, as they can become effectors for chronic inflammation, impair wound healing, and cause tissue damage and organ failure.

Nevertheless, information on the positive effects of NETs, other than their antimicrobial ability, although limited in number, are promising. They could become useful therapeutic tools in the future.

## Figures and Tables

**Figure 1 cells-10-00494-f001:**
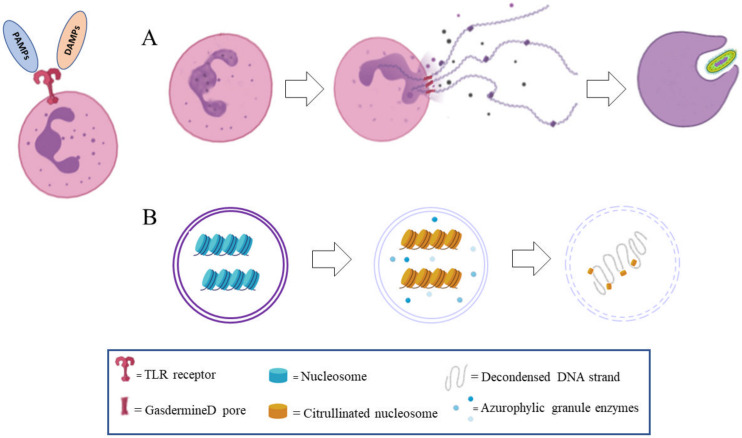
Molecular mechanism involved in the neutrophils extracellular traps (NETs) formation process. Following pathogen-associated molecular patterns (PAMPs) or damage-associated molecular patterns (DAMPs), molecules binding to toll-like receptors (TLR) in a neutrophil granulocyte several steps characterizing the evolution, formation, and extrusion of NETs. (**A**) Cytoplasmatic events: cytoplasm granules are shifted into the nucleus for decondensing DNA, the nucleus loses its integrity, and NETs meshwork is released in the extracellular space by gasdermin D pores. After NETs extrusion, the granulocyte preserves its phagocytic ability for some hours. (**B**) Nuclear events: nuclear DNA-histones is was decondensed by azurophilic granulocyte enzymes, then linker H1 histone is cleaved off and histone 3 and 4 are citrullinated, inducing DNA decondensation.

**Figure 2 cells-10-00494-f002:**
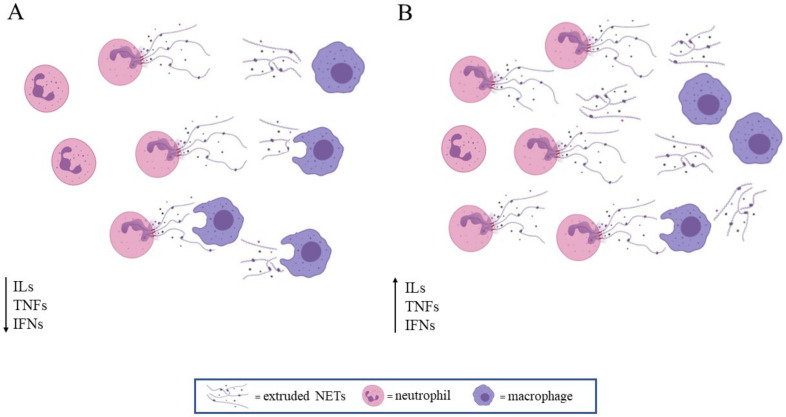
NETosis in physiologic (**A**) and pathologic conditions or ageing (**B**). Interleukins (ILs), tumor necrosis factors (TNFs), and interferones (IFNs). (**A**) NETs extrusion is adequate to the defensive anti-microbial action, and NETs are then removed by macrophages; inflammatory factors drop down. (**B**) Basal or induced NETs extrusion is enhanced, and they are not efficiently removed by macrophages; inflammatory factors increase, leading to long-lasting inflammation.

**Figure 3 cells-10-00494-f003:**
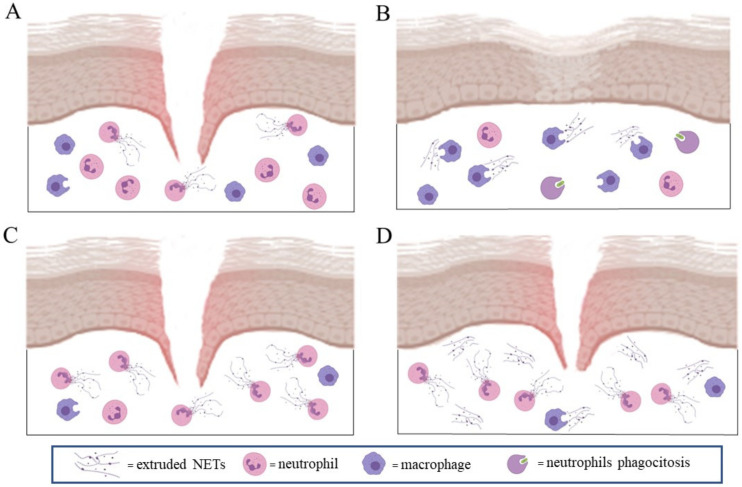
NETs in wound healing: the physiological release of NETs and macrophage clearance (**A**) actions are essential to support the physiological response of fibroblasts and keratinocytes to heal wounds (**B**). Excessive NETs extrusion and inefficient macrophage clearance (**C**) increases in situ inflammation and altered both fibroblast and keratinocyte pro-healing action (**D**).

## Data Availability

Not applicable.

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
