# Peer review of "NETosis in Wound Healing: When Enough Is Enough"

_cells, 2021, doi:10.3390/cells10030494_

Round 1

Reviewer 1 Report

The manuscript in terms of English grammar is improved but several errors remain.  Examples are listed below:

Page 4, first paragraph.  "NETS have also been observed to be efficacious in the host defense..."

Page 4, paragraph 3.  "...microbes limiting their diffusion and concentrating (?) neutrophil factors..."

Page 4, paragraph 4.  "....chelating activity of the DNA in the meshwork of NETS decreased ion transport is impeding the survival..."

Page 5, paragraph 1.  "Finally, the release of chromatin treads granule and factors takes place...."  This sentence is incomprehensible.

Page 7, paragraph 1.  "...there is an activation of the immune system response which clears dead cells triggering inflammation and is necessary..."

Page 7, paragraph 2.  "...leading them to package and activate the NADPH..."

Page 7, paragraph 2.  "...probably by hypochlorite synthesis.  In fact subjects with an altered MPO gene..."

Page 8, paragraph 5.  "...where lungs are the main target, acute lung injury (ALI) and its most severe form, the acute respiratory distress syndrome (ARDS)."

Page 10, paragraph 3.  "In particular, in a murine breast cancer model, NETs promote the progression of tumor metastasis due to protein factors..."

Page 11, paragraph 1.  "These events can induce a delay in the macrophage dependent downregulation of inflammation causing an inappropriate persistence of autoantigen responsible for the aggravated condition..."

Page 11, paragraph 4. "In the case of neutrophil recruitment to the area of inflammation induces the release..." "...by binding and sequestering inflammatory cytokines such as IL-1B and IL-6, which are then degraded by serine..."

Page 12, paragraph 3.  "...increased NETosis has been shown to impair wound healing..." "....healing due to NET formation has been well reported in diabetic patients..."

Page 13, paragraph 4.  "NETosis dysregulation can also be induced by chronic inflammation, a mechanism responsible for the delayed..."

Page 14, paragraph 2.  "IL-17 is a proinflammatory cytokine which has been shown to be associated with an increase and propagation of a fibrotic process in several different tissues including lung, skin, liver, and others..."

Page 15, paragraph 3.  "Also, keratinocytes express TLR9, with its activation inducing type IINF release."

Page 16, paragraph 2.  "...where less NET production was observed in neutrophils obtained from elderly animals compared to those obtained from younger animals."

Page 17, paragraph 1.  "Type 2 diabetes mellitus (T2D) patients are known to be at an increased risk for infectious and wound healing impairment, which have been partially linked..."

Page 17, paragraph 2.  "...cathelicidin (LL-37) that possess poor antimicrobial..."

Page 17, paragraph 4.  "Nevertheless, information on the bright side of NETs, other than their antimicrobial ability, although limited in number, are promising as they could become..."

In several places, the pleural form of words is used incorrectly.  For example on Page 8, line 3 "protein(s)", Page 8, line 5, neutrophil(s)

Author Response

All the errors indicated by the reviewer have been corrected and the corrections are highlighted in the text

Reviewer 2 Report

  It is an interesting review article describing neutrophils extracellular traps (NET) formation, a neutrophils physiological response to regulate inflammation. The authors summarized recent evidence suggesting that there are positive NET effects on wound healing with the therapeutic potentials.

  The manuscript is well written in English and the contents are relevant to the clinical application. There are some suggestions as follows.

  1. In this manuscript, general description on NET has 9 pages (page 3 to 12), but the length related to wound healing is only 5 pages (page 12 to 17). The authors should consider to revise the manuscript title as “NETosis and its role in wound healing – “.
  2. There were some typographical errors in the manuscript. For example, on page 4 line 5, “be efficient in” rather than “be efficacy in”. On page 8 line 6, “is also involved” rather than “also is involved”.
  3. In Figure 2, the authors should show the full names of ILs, TNFs and IFNs in figure legends. Similar to Figure 1, in Figures 2 and 3, the authors should describe the illustrated patterns in these figures such as decondensed DNA. 

Author Response

The authors thank the reviewer for the interesting suggestions:

  1. The manuscript has been submitted to a special issue on Wound Healing…, that’s why we have chosen this title.
  2. All the typographical errors have been corrected and highlighted.
  3. Figures 2 and 3 have been modified as requested.

Reviewer 3 Report

Comments to authors

The article addresses a major problem in the field of neutrophil biology.

Abstract: The phrase "The positive NETs of effects on wound healing.." do not sounds correct. Perhaps it is more corrected "The positive effects of NETs on wound healing.."

What do you means with phrase "..the hypothesis that there are other positive physiological and positive NETs effects on wound healing"?

Neutrophil kinetics studies show that intravascular half- life of neutrophils is longer than we believed, but reference that you used (Scapini, P.; Cassatella, M.A. Social networking of human neutrophils within the immune system. Blood 2014, 124, 710–9) do not fully supports these findings. In my opinion, the article "The Neutrophil Life Cycle" by Hidalgo et al., is more suitable.

The paragraph "NETosis dark side" has been written confusingly. You start with description of NETs' role in chronic diseases like atherosclerosis, diabetes cancer, but after reference 37 jump to acute lung injury. To continuous the paragraph with COVID, you give emphasis on modern trends, but not describe the negative role neutrophils via NETs in chronic disorders.

The role of NETs in COVID may re-written in separate paragraph.

The paragraph "NETosis bright side" starts with debated dilemma.

There are many articles that introduces the NETs into gout pathogenesis. The role of proinflammatory NETs and IL-1β in pathogenesis of gout was document for the first time by Mitroulis et al. Indeed, Hahn et al, in their review article hypothesized that neutrophils, via NETs formation, play role in resolution of inflammation, but neutrophils are not enough for resolution of acute gouty arthritis as shown by Reber et al.

Page 16: please write in vitro and microbial names (Staph aureus) on italics. The paragraph "This NET concentration-dependent effect.." is sequel of previous paragraph.

Author Response

Abstract: the two phrases indicated have been corrected.

The paper by Scapini et al., 2014 has been substitutes by the paper by Hidalgo et al., 2019.

The paragraph “NETosis dark side” has been reorganized, but the role of NETs in COVID has not been re-written in a separate paragraph, just because the subject is not a key topic of our paper.

The paragraph “NETosis bright side” has been reorganize including the new references Reber et al. 2016, and Mitroulis et al., 2013.

Pag. 16, all the suggestion has been accepted.

This manuscript is a resubmission of an earlier submission. The following is a list of the peer review reports and author responses from that submission.

Round 1

Reviewer 1 Report

Sabbatini et al offer a review of NETosis in wound healing in order to address the double edged sword of NET release on tissue damage vs. tissue repair. A strength of the review is that it is built upon 112 references which are appropriate to cover the topic and include many primary papers published within the past five years supporting the timeliness of the work. However there are a number of concerns regarding accuracy and preparation:

  1. The manuscript is very poorly prepared. The authors must work with an English speaking editor because most of their sentences are grammatically awkward and sometimes used words that do not exist in the English language. Also, there are examples of non-sequiturs which makes the point the authors intend to make difficult to understand. One example occurs in the very first paragraph which refers to neutrophil life span which does not follow in a logical sequence with NET activation.

Several issues are seen on Page 2; paragraph 5:

“Therefore, the difference between a “vital NETosis” and a “lethal NETosis” such as was previously pronounced and discussed have no more sense. {This sentence makes little grammatical sense} Under this point of view some researcher have criticized the use of the term “NETosis entirely preferring the term NET formation [18]. NET formation seems to be a diffuse phenomenon inside responses of immunitary cells, and aside from neutrophils, otherimmunitary cells also were reported develop the process of NETosis [7,19-21]…{Immunitary is not a word}

  1. Whereas this reviewer did not engage in checking many references, paragraph five refers to the original work by Brinkmann et al and indicates that NETs were induced by enveloped viruses which is not covered in that paper. (see PLoS One 2011;6(7):e22043.doi: 10.1371/journal.pone.0022043). However the sentence also serves as one aforementioned example of the need for editing because the sentence ends with “…enveloped viruses also: neutrophil extracellular traps (NETs)” which does not make grammatical sense.

Another questionable referencing issue is in paragraph 6 “NETs are released in the extracellular space where the chromatin meshwork entraps microbes limiting their diffusion and gathering the neutrophil factors, increasing the microbicidal effects [7-9]. NET release became fundamental as defensive mechanism, when pathogens dimension makes phagocytosis a not reliable process [10].”

            Here reference 10 should follow the first sentence and the work that supports the latter sentence is Nat Immunol 2014 Nov;15(11):1017-25.doi: 10.1038/ni.2987. 

Reference 60 is redundant with reference 78. Here again I did not scour the reference list for any further examples but cite this one as suggestive that there may be others and the manuscript is not carefully prepared.

Reference 9 is incomplete and the phrase “Beta-glucan that are part of bacteria” to which the reference is given in support is not correct as glucan is a fungal PAMP.

Author Response

REVIEWER 1

1.Authors apologize for the linguistic and grammar mistakes, the manuscript has been completely  revised by a linguistic point of view.

Non-sequitur sentences have been modified or eliminated.

Page 2, paragraph  5:

  • Non sequitur sentence has been rephrased in a more consequential presentation.
  • The sentence about vital and lethal NETosis has been deleted.
  • The expression “immunitary cells” has been corrected.

2.

  • The indication about Brinkmann paper has been corrected and reported in a more appropriate way;
  • The appropriate citation indicated by reviewer on “enveloped viruses” has been inserted;
  • The entire paragraph has been rephrased and more specific bibliographic references have been inserted

Page 2, paragraph  6:

  • Citation has been corrected according to reviewer’s suggestion.

- Reference 78 has been deleted.

- Reference 9 has been deleted and substitute with a more specific citation on PAMP and DAMP.

- Indication on β-glucan has been corrected, a more appropriate citation has been inserted.

- All references reported in the manuscript have been carefully checked.

Reviewer 2 Report

The manuscript by Sabbatani, et al entitled “NETosis in wound healing: when enough is enough” reviews an important and developing topic.  Since their time of initial identification and description, the role of NETs in acute inflammation, both infectious and non-infectious, has expanded.  This review nicely describes and discusses the origins of NETs and some of the pathways regulating their production.  However, although the title of the review suggests that the role of NETs on wound healing will be extensively discussed, only a small fraction of the review focusses on NETs and wound healing.  This area of the review is also poorly organized.  As chronic inflammatory diseases (e.g. psoriasis) are also dependent on wound healing for clinical improvement, the earlier discussion on the role of NETs in psoriasis would be better placed in the section on wound healing.  As a focus on NETs, or the pathways regulating their formation and release, may be a therapeutic target for the treatment of wounds in the future, a detailed and organized discussion regarding the positive and negative effects of NETs on wound healing would be welcomed by the research community.  This review begins that process but needs to be improved to reach its goal.

Throughout the manuscript there are errors in grammar that would benefit from review by a native English speaker.  These errors make the manuscript difficult to read in places.  

Author Response

REVIEWER 2

Authors apologize for the linguistic and grammar mistakes, the manuscript has been completely revised by a linguistic point of view.

The section of wound healing has been extended and better organized.

The discussion about NETs in psoriasis has been shifted in the wound healing section as suggested by reviewer.

A more detailed and organized discussion regarding the positive and negative effect of NETs on wound healing has been organized.

Reviewer 3 Report

Sabbatini M et al. present a comprehensive review on the healing properties of NETs together with the latest knowledge on the topic. It would be interesting to briefly discuss the role of IL-17 and antimicrobial peptide Cathelicidin (LL-37) on fibrosis and wound healing in diabetes and SLE.

Frangou E, et al. REDD1/autophagy pathway promotes thromboinflammation and fibrosis in human systemic lupus erythematosus (SLE) through NETs decorated with tissue factor (TF) and interleukin-17A (IL-17A). Ann Rheum Dis. 2019 Feb;78(2):238-248.

Arampatzioglou A, et al. Clarithromycin Enhances the Antibacterial Activity and Wound Healing Capacity in Type 2 Diabetes Mellitus by Increasing LL-37 Load on Neutrophil Extracellular Traps. Front Immunol. 2018 Sep 10;9:2064.

Author Response

REVIEWER 3

 A discussion about the role of IL-17 and LL-37 on fibrosis and wound healing in diabetes and SLE has been introduced in the text.

Round 2

Reviewer 1 Report

The issues I listed were only a few representative problems which would otherwise be too numerous to list in a review. My recommendation still stands.

Reviewer 2 Report

No additional comments